# Genome-Wide Analysis of Aquaporins in Japanese Morning Glory (*Ipomoea nil*)

**DOI:** 10.3390/plants12071511

**Published:** 2023-03-30

**Authors:** Tamami Inden, Atsushi Hoshino, Shungo Otagaki, Shogo Matsumoto, Katsuhiro Shiratake

**Affiliations:** 1Graduate School of Bioagricultural Sciences, Nagoya University, Nagoya 464-8601, Japan; 2National Institute for Basic Biology, Okazaki 444-8585, Japan; 3Department of Basic Biology, School of Life Science, SOKENDAI, Okazaki 444-8585, Japan

**Keywords:** genome-wide analysis, morning glory, aquaporin, AQP, *Ipomoea nil*

## Abstract

The aquaporin (AQP) family, also called water channels or major intrinsic proteins, facilitate water transport. AQPs also transport low-molecular-weight solutes, including boric acid, glycerol, urea, and ammonia. Since plants are sessile, water homeostasis is crucial. Therefore, plants have developed diverse AQP variants at higher expression levels than animals. For example, 35 and 33 AQPs have been identified in *Arabidopsis* and rice, respectively. In the present study, we identified AQPs in morning glory (*Ipomoea nil*), which has been widely used as a model plant in research on flowering and floral morphology. The importance of AQPs in the opening of morning glory flowers has been reported. In the morning glory genome, 44 AQPs were identified, and their characteristics were analyzed. A phylogenetic analysis revealed five AQP subfamilies in morning glory: plasma membrane-intrinsic proteins (PIPs), tonoplast-intrinsic proteins (TIPs), nodulin 26-like intrinsic proteins (NIPs), small basic intrinsic proteins (SIPs), and X-intrinsic proteins (XIPs). Further, transport substrates of morning glory AQPs were estimated based on their homology to the known AQPs in other plant species and their corresponding amino acid motifs that possess permeability pores. It was expected that PIPs are likely to transport water, carbon dioxide, and hydrogen peroxide; TIPs are likely transport water, hydrogen peroxide, ammonia, urea, and boric acid; NIPs are likely transport water, boric acid, ammonia, glycerol, and formamide; and XIPs are likely to transport water, hydrogen peroxide, and glycerol. Overall, these results suggest that AQPs are involved in water and nutrient transport in Japanese morning glory. An in silico gene expression analysis suggested the importance of AQPs in flower opening, water or nutrient uptakes from the soil to roots, and photosynthesis in morning glory. Our findings provide fundamental information that enables further study into the importance of AQPs in morning glory, including their roles in flower opening and other physiological events.

## 1. Introduction

Water is essential for plant growth and various other physiological events, including photosynthesis. Plants maintain cellular functions and growth by absorbing water into cells by taking up water from the roots and transporting it to various organs. Moreover, water transport is important for the distribution of nutrients and photoassimilates in plants. Unlike animals, plants are unable to search for water. Therefore, they have developed mechanisms to maintain water homeostasis and adapted to different water conditions. In these processes, aquaporins (AQPs) play indispensable roles; consequently, plants have developed a large gene family of AQPs to adapt to various water conditions.

AQPs are membrane proteins that transport many water molecules with high selectivity. AQPs exhibit the fastest transport speed among various transporters, including ion channels, and they can transport over one billion water molecules per second under the appropriate conditions [1]. In addition to water, AQPs transport small neutral molecules, such as glycerol, urea, ammonia, carbon dioxide, hydrogen peroxide, and boric acid [2,3,4,5].

AQPs are composed of several conserved and characteristic structures, including six transmembrane domains (TMDs) and five connecting loops (loops A–E). Loop B and loop E form a half-helix, and inside each of the half-helices, two asparagine–proline–alanine (NPA) motifs are positioned facing each other (Figure 1). The NPA motifs form a pore with a diameter of 3 Å, through which water molecules permeate. The ar/R filter, comprising four amino acid residues, prevents the passage of molecules other than the substrate; therefore, this structure is important for substrate selectivity. Meanwhile, the Froger’s position, comprising five amino acid residues, has been postulated to selectively transport substrates other than water [6].

AQPs are universal in all organisms, including bacteria, plants, and animals [7,8,9]. Thirteen AQPs are present in humans [10], 35 in *Arabidopsis* [11], 33 in rice [12], and 47 in tomato [13]. AQP expression is higher in plants than in animals, indicating their importance and diverse roles in the former. As previously discussed, plants cannot move in the search for water. Therefore, they possess a highly developed water transport system with diverse AQP families. AQPs in higher plants are classified into five subfamilies based on homology: plasma membrane-intrinsic proteins (PIPs), tonoplast-intrinsic proteins (TIPs), nodulin 26-like intrinsic proteins (NIPs), small basic intrinsic proteins (SIPs), and X-intrinsic proteins (XIPs) [14,15].

Morning glory (*Ipomoea nil*) is cultivated as an ornamental plant in Japan. In addition, it is used as a model plant in flower research because of the high sensitivity of its flowering to day length, alongside the presence of various mutants of flower colors and the peculiar morphologies of flowers and other organs. These mutants are collected and distributed by the National BioResource Project (NBRP) morning glory (https://shigen.nig.ac.jp/asagao/index.jsp (accessed on 20 August 2021)). The whole genome of Japanese morning glory was sequenced by Hoshino et al. (2016) [16].

AQPs play pivotal roles in the opening and closing of the flowers of plants [17,18], including morning glory [1,19]. Therefore, in the present study, we performed a genome-wide analysis of AQPs in morning glory to predict their corresponding roles. First, the genes encoding AQPs were identified in the morning glory genome. Then, their gene and protein sequences predicted subcellular localization, predicted substrates, and then the gene expression profiles were annotated. Furthermore, based on this information, the putative roles of AQPs in morning glory were discussed.

## 2. Results

### 2.1. Genome-Wide Analysis of AQPs in Japanese Morning Glory

To identify AQPs in morning glory, BLASTP was used to search against the Japanese morning glory genome database. Specifically, a BLASTP search was conducted using the amino acid sequences of *Arabidopsis* and tomato AQPs [11,13] as queries. In total, 45 AQP candidates were identified (Appendix A). The annotation of the best hit homolog in NCBI was AQP, and the numbers of amino acid residues in the candidates were similar to those in AQPs from other plant species (100–350 residues). An analysis of genes encoding the 45 AQP candidates revealed that INIL04g32909 and INIL04g32910 were identical to XM_019307143.1 in the Gnomon annotation (Appendix A). In addition, mapping of the RNA-Seq reads demonstrated that INIL04g32909 lacked RNA-Seq reads in most parts of the second exon; nonetheless, the reads of the coding region of INIL04g32909 still matched the coding region of XM_019307143.1 (Appendix A). Therefore, we concluded that INIL04g32909 and INIL04g32910 represented the same gene and used the Gnomon annotation XM_019307143.1 (InSIP1;1), which was implemented in further analyses. Finally, we identified 44 AQP genes in the Japanese morning glory genome (Table 1).

In a phylogenetic analysis based on the amino acid sequences of the candidate morning glory, *Arabidopsis* and tomato AQPs, the forty-four morning glory AQPs were classified into five families: thirteen PIPs, eleven TIPs, fifteen NIPs, two SIPs, and three XIPs (Figure 2 and Appendix A). These families were further classified into subfamilies, and the morning glory AQPs were named according to the order of the gene ID (chromosome number and position on the chromosome; Table 1, Figure 2). The InPIPs were classified into four InPIP1s and nine InPIP2s. The InTIPs were classified into six InTIP1s, two InTIP2s, one InTIP3, one InTIP4, and 1 one (Figure 2). The sequence of INIL03g01865 (InTIP3;1) contained an unknown sequence X, which was absent in the corresponding gene in the Gnomon annotation (XM_019346257.1). Furthermore, its RNA-Seq read matched the entire XM_019346257.1 sequence (Appendix A). Therefore, the sequence of XM_019346257 was used as the InTIP3;1 sequence in further analyses. InNIPs were divided into seven subfamilies: three InNIP1s, three InNIP2s, one InNIP3, three InNIP4s, three InNIP5s, one InNIP6, and one InNIP7 (Figure 2). A phylogenetic analysis indicated that INIL06g37831 (InNIP4;2) was highly similar to tomato SlNIP2;2 (Appendix A), although its ar/R filter, which is important for substrate selectivity, was similar to that of NIP4s (Table 2). Therefore, INIL06g37831 was classified as an InNIP4. INIL08g31098 (InNIP4;3) and INIL06g37646 (InNIP5;2) carried deletions in their amino acid sequences, and the second half of the AQP motif sequence was absent, which had no deletions; the AQP motif sequence was present in the corresponding Gnomon annotations (XM_019304577.1 and XM_019314787.1, respectively). Therefore, the sequences of XM_019304577.1 and XM_019314787.1 were used as InNIP4;3 and InNIP5;2, respectively, in further analyses.

### 2.2. Exon–Intron Structure

The gene structures of morning glory AQPs were schematized based on their genome and coding sequences (CDSs) using the Gene Structure Display Server 2.0. Most morning glory AQP genes carry 2–6 exons, and the corresponding gene structures vary across AQP families (Figure 3). Most InPIPs possess three or four exons; most InTIPs possess three exons; most InNIPs possess four or five exons; InSIPs possess three exons; and InXIPs possess two or three exons.

### 2.3. TMDs

Typical plant AQPs comprise 200–300 amino acid residues and six TMDs [20] (Figure 1). Here, the TMDs of morning glory AQPs were predicted using two different prediction programs, TMHMM and SOSUI (Table 1). Predictions using both programs were generally consistent, although there were differences in some predicted TMDs. Among the 44 morning glory AQPs, 40 AQPs, except InNIP2;3, InNIP5;3, InSIP1;1, and InSIP2;1, were predicted to have six TMDs by TMHMM or SOSUI (Table 1). The alignment of InNIP2;3 with other NIPs suggested that InNIP2;3 lacks the amino acid sequence from the second NPA motif in the sixth TMD (Appendix A). Although no amino acid deletions were detected in InNIP5;3, InSIP1;1, and InSIP2;1, the corresponding numbers of TMDs were four, five, and seven, respectively (Table 1). Large amino acid sequence deletions in AQPs, such as InNIP2;3, may be caused by the misprediction of their gene structure.

**Table 1 plants-12-01511-t001:** **AQP genes identified in the Japanese morning glory genome.**

Name	Gene ID		Location			Amino Acid No.	Molecular Weight (kDa)	TMDs ^◇^		pI ^§^	Subcellular Localization	
	AUGUSTUS	Gnomon (NCBI)	Chromosome No.	Start	End			TMHMM	SOSUI		Plant-mPLoc	Wolf PSORT
**InPIP1;1**	INIL02g16994	XM_019337031.1	chr2	41500196	41501995	284	30.4	6	5	8.8	cell membrane	plas: 9, cyto: 3, pero: 1
**InPIP1;2**	INIL05g23971	XM_019295874.1	chr5	35829017	35831432	287	31.0	6	6	9.0	cell membrane	plas: 8, cyto: 3, pero: 2
**InPIP1;3**	INIL05g24206	XM_019295755.1	chr5	39507889	39510158	285	30.4	6	5	7.7	cell membrane	plas: 10, cysk: 2, cyto: 1
**InPIP1;4**	INIL09g29995	XM_019303783.1	chr9	197300	198859	285	30.4	6	6	8.8	cell membrane	plas: 7, cyto: 3, cysk: 2, vacu: 1
**InPIP2;1**	INIL03g18022	XM_019337682.1 XM_019337681.1	chr3	36083595	36085245	284	30.3	6	5	9.0	cell membrane	plas: 11, golg: 2
**InPIP2;2**	INIL04g34732	XM_019309792.1 ^△^	chr4	273291	275487	289	30.8	6	6	9.1	cell membrane	plas: 8.5, cyto_plas: 5, E.R.: 2, chlo: 1, mito: 1
**InPIP2;3**	INIL05g09496	XM_019327277.1	chr5	1980051	1982031	270	28.5	6	4	9.1	cell membrane	plas: 11, vacu: 2
**InPIP2;4**	INIL08g34668	XM_019311043.1	chr8	29166734	29169262	282	30.2	6	5	8.8	cell membrane	plas: 10, cysk: 3
**InPIP2;5**	INIL09g21386	XM_019343259.1 XM_019343260.1	chr9	4620360	4622484	282	30.0	6	5	7.7	cell membrane	plas: 9, cysk: 3.5, cysk_nucl: 2.5
**InPIP2;6**	INIL10g11305	XM_019329230.1	chr10	35274307	35275867	284	30.3	6	5	7.6	cell membrane	plas: 12, cyto: 1
**InPIP2;7**	INIL12g15802	XM_019335920.1	chr12	5285444	5286882	285	30.3	6	5	8.5	cell membrane	plas: 12, golg: 2
**InPIP2;8**	INIL12g15803	XM_019335914.1	chr12	5277886	5279235	285	30.3	6	5	8.5	cell membrane	plas: 13
**InPIP2;9**	INIL12g22089	XM_019343855.1	chr12	60062545	60065025	287	30.8	6	5	7.7	cell membrane	plas: 11, cysk: 3
**InTIP1;1**	INIL02g16891	XM_019336315.1	chr2	40786840	40788260	252	25.8	6	6	5.8	vacuole	vacu: 6, plas: 3, chlo: 2, cyto: 2
**InTIP1;2**	INIL04g02479	XM_019296873.1	chr4	12385183	12386743	246	24.9	6	6	6.3	vacuole	cyto: 9, vacu: 4
**InTIP1;3**	INIL07g08709	XM_019326170.1	chr7	14480111	14481905	245	25.1	6	6	6.1	vacuole	vacu: 6, cyto: 4, plas: 4
**InTIP1;4**	INIL08g00150	XM_019331552.1	chr8	26850052	26851598	251	25.9	4	5	6.0	vacuole	plas: 4, vacu: 4, cysk: 3, chlo: 1, cyto: 1
**InTIP1;5**	INIL11g18467	XM_019337154.1	chr11	24689421	24690925	249	25.7	6	6	5.9	vacuole	plas: 6, vacu: 5, cyto: 2
**InTIP1;6**	INIL12g22155	XM_019343462.1	chr12	60755671	60756940	252	25.9	6	7	5.6	vacuole	plas: 5, vacu: 5, cyto: 2, chlo: 1
**InTIP2;1**	INIL01g19987	XM_019341113.1	chr1	35538537	35540087	248	25.1	7	6	5.6	vacuole	vacu: 6, cyto: 4, plas: 4
**InTIP2;2**	INIL03g18342	XM_019338205.1 XM_019338204.1	chr3	3827838	38279749	248	25.1	7	6	5.6	vacuole	plas: 6, vacu: 6, chlo: 1
**InTIP3;1 ***	INIL03g01865	XM_019340257.1	chr3	444425	445682	262	27.7	6	6	6.8	vacuole	mito: 6, cyto: 2, plas: 2, vacu: 2, nucl: 1
**InTIP4;1**	INIL14g06843	XM_019323579.1	chr14	39466956	39469705	247	25.6	7	6	5.8	vacuole	plas: 7, vacu: 4, cyto: 2
**InTIP5;1**	INIL02g17136	XM_019336883.1 XM_019336884.1	chr2	42529270	42530437	253	26.2	6	6	6.0	cell membrane and vacuole	Cyto: 7, chlo: 2.5, vacu: 2, chlo_mito: 2, nucl: 1
**InNIP1;1**	INIL06g35686	XM_019311095.1	chr6	14388531	14392215	276	29.3	6	6	8.9	cell membrane	plas: 8, vacu: 3, cysk: 2
**InNIP1;2**	INIL14g02021	XM_019343044.1	chr14	8114134	8117138	249	26.2	5	6	5.9	cell membrane	plas: 12, E.R.: 2
**InNIP1;3**	INIL14g02022	XM_019343057.1	chr14	8098734	8100242	256	27.1	6	6	6.5	cell membrane	plas: 12, vacu: 1
**InNIP2;1**	INIL00g00315 ^#^	XM_019318254.1	scaffold0125	6546	9471	292	31.1	6	6	9.3	cell membrane	plas: 10, golg: 3
**InNIP2;2**	INIL00g18375 ^#^	XM_019339178.1	scaffold1424	4008	6927	314	34.2	5	5	9.6	cell membrane	plas: 9, E.R.: 2, golg: 2
**InNIP2;3**	INIL11g18696	XM_019339456.1	chr11	8328345	8331219	247	26.8	4	4	9.8	cell membrane	plas: 7.5, cyto_plas: 6, cyto: 3.5, E.R.: 2
**InNIP3;1**	INIL14g41661	XM_019319645.1	chr14	56617068	56618733	266	28.9	5	6	6.4	cell membrane	plas: 10, E.R.: 2, vacu: 1
**InNIP4;1**	INIL02g40633	XM_019317591.1	chr2	8659937	8663157	261	27.8	6	6	7.7	cell membrane	plas: 9, vacu: 3, nucl: 1
**InNIP4;2**	INIL06g37831	XM_019313478.1	chr6	4421259	4423701	295	31.9	6	6	6.1	cell membrane	plas: 7.5, cyto_plas: 4.5, golg: 3, extr: 1, vacu: 1
**InNIP4;3 ***	INIL08g31098	XM_019304577.1	chr8	6161219	6162577	270	29.0	6	7	8.3	cell membrane	plas: 7, nucl: 3, cyto: 2, chlo: 1
**InNIP5;1**	INIL05g21764	XM_019343441.1	chr5	27557396	27563220	296	30.8	5	6	8.3	cell membrane	plas: 7, vacu: 5, extr: 1
**InNIP5;2 ***	INIL06g37646	XM_019314787.1	chr6	6126438	6128483	300	31.3	5	6	6.6	cell membrane	plas: 11, vacu: 2
**InNIP5;3**	INIL15g31300	XM_019304934.1	chr15	7534705	7538310	250	25.5	5	soluble	9.0	cell membrane	plas: 6, cyto: 4, vacu: 3
**InNIP6;1**	INIL09g36261	XM_019311905.1	chr9	8779488	8782317	220	22.8	6	5	9.0	cell membrane	plas: 6, vacu: 6, nucl: 1
**InNIP7;1**	INIL13g08057	XM_019325163.1 XM_019325162.1	chr13	2555745	2559812	235	25.0	6	6	6.4	cell membrane and vacuole	Vacu: 10, plas: 3
**InSIP1;1 ^○^**	INIL04g32909	XM_019307143.1	chr4	7444038	7444872	239	25.3	5	7	9.6	cell membrane and vacuole	Vacu: 5, chlo: 2, cyto: 2, extr: 2, nucl: 1, mito: 1
	+											
	INIL04g32910		chr4	7444938	7446809							
**InSIP2;1**	INIL04g32936	XM_019307287.1	chr4	7664318	7666212	241	26.5	4	4	9.1	cell membrane	vacu: 7, plas: 3, E.R.: 3
**InXIP1;1**	INIL04g34767	XM_019309584.1	chr4	432408	433612	321	34.8	6	6	7.0	cell membrane	plas: 10, golg: 3
**InXIP1;2**	INIL06g38432	XM_019313788.1	chr6	308805	310475	339	36.3	6	7	7.7	cell membrane	plas: 9.5, cyto_plas: 5.5, E.R.: 2, vacu: 1
**InXIP1;3**	INIL06g38434	XM_019313789.1 XM_019313790.1	chr6	301085	302502	335	35.8	6	6	7.2	cell membrane	plas: 8.5, cyto_plas: 5, E.R.: 2, golg: 2

* Since the amino acid sequence of INIL03g01865 contained X and INIL08g31098 and INIL06g37646 had deletions in the amino acid sequence, the molecular weight, transmembrane domains, isoelectric point, and subcellular localization were predicted using the amino acid sequence of Gnomon annotations; # the chromosome number is unknown; therefore, scaffold is shown; ^◇^ transmembrane domains; § isoelectric point; ^○^ InSIP1;1 is expected to consist of two genes (INIL04g32909, INIL04g32910); ^△^ and this gene is expected to encode something other than an AQP; therefore, this is considered to be an annotation error.

### 2.4. Conserved Motifs and Subcellular Localization

Plant AQPs possess several conserved and characteristic protein motifs, including two highly conserved NPA motifs, the ar/R filter (H2, H5, LE1, and LE2), and the Froger’s position (P1–P5) (Figure 1). These motifs are important for the selectivity of the substrate transport [2,6]. These motifs in morning glory AQPs were confirmed in this study and are summarized in Table 2.

**Table 2 plants-12-01511-t002:** **Amino acid composition of the NPA motifs, ar/R filters (H2, H5, LE1, and LE2), and Froger’s positions (P1–P5) in Japanese morning glory AQPs.**

Name	NPA Motif ^1^		ar/R Filter ^2^				Froger’s Positions ^3^				
	First	Second	H2	H5	LE1	LE2	P1	P2	P3	P4	P5
InPIP1;1	NPA	NPA	F	H	T	R	M	S	A	F	W
InPIP1;2	NPA	NPA	F	H	T	R	M	S	A	F	W
InPIP1;3	NPA	NPA	F	H	T	R	M	S	A	F	W
InPIP1;4	NPA	NPA	F	H	T	R	M	S	A	F	W
InPIP2;1	NPA	NPA	F	H	T	R	A	S	A	F	W
InPIP2;2	NPA	NPA	F	H	T	R	A	S	A	F	W
InPIP2;3	NPA	NPA	F	H	T	R	A	S	A	F	W
InPIP2;4	NPA	NPA	F	H	T	R	A	S	A	F	W
InPIP2;5	NPA	NPA	F	H	T	R	A	S	A	F	W
InPIP2;6	NPA	NPA	F	H	T	R	A	S	A	F	W
InPIP2;7	NPA	NPA	F	H	T	R	A	S	A	F	W
InPIP2;8	NPA	NPA	F	H	T	R	A	S	A	F	W
InPIP2;9	NPA	NPA	F	H	T	R	A	S	A	F	W
InTIP1;1	NPA	NPA	H	I	A	V	T	S	A	Y	W
InTIP1;2	NPA	NPA	H	I	A	V	T	A	A	Y	W
InTIP1;3	NPA	NPA	H	I	A	V	T	A	S	Y	W
InTIP1;4	NPA	NPA	H	I	A	V	T	S	A	Y	W
InTIP1;5	NPA	NPA	H	I	A	V	T	A	A	Y	W
InTIP1;6	NPA	NPA	H	I	A	V	T	S	A	Y	W
InTIP2;1	NPA	NPA	H	I	G	R	T	S	A	Y	W
InTIP2;2	NPA	NPA	H	I	G	R	T	S	A	Y	W
InTIP3;1	NPA	NPA	H	V	A	R	L	A	A	Y	W
InTIP4;1	NPA	NPA	H	I	A	R	T	S	A	Y	W
InTIP5;1	NPA	NPA	N	V	G	Y	T	A	A	Y	W
InNIP1;1	NPA	NPA	W	V	A	R	F	S	A	Y	L
InNIP1;2	NPA	NPA	W	V	A	R	F	S	A	Y	M
InNIP1;3	NPA	NPA	W	V	A	R	F	S	A	Y	M
InNIP2;1	NPA	NPA	G	I	-	R	L	T	A	Y	I
InNIP2;2	NPA	NPA	G	S	G	R	L	T	A	Y	I
InNIP2;3	NPA	-	G	S	-	-	L	-	-	-	-
InNIP3;1	NPA	NPA	W	V	A	R	F	S	A	F	V
InNIP4;1	NPA	NPA	W	V	A	R	F	S	A	Y	I
InNIP4;2	NPA	NPA	W	S	A	R	F	T	A	Y	M
InNIP4;3	NPA	NPA	W	V	A	R	L	T	A	Y	L
InNIP5;1	NPS	NPV	A	I	G	R	F	T	A	Y	L
InNIP5;2	NPS	NPV	A	I	G	R	F	T	A	Y	L
InNIP5;3	NPS	NPV	A	I	G	R	F	S	A	Y	L
InNIP6;1	NPA	NPV	T	I	A	R	F	T	A	Y	L
InNIP7;1	NPS	NPA	A	V	A	R	Y	S	A	Y	F
InSIP1;1	NPT	NPA	I	T	P	N	F	A	A	Y	W
InSIP2;1	NPL	NPA	F	K	G	S	I	V	A	Y	W
InXIP1;1	NPI	NPA	I	T	A	R	V	C	A	F	W
InXIP1;2	NPT	NPA	I	T	A	R	V	C	A	F	W
InXIP1;3	NPI	NPA	I	T	A	R	V	C	A	F	W

^1^ Asparagine–proline–alanine motifs in loops B (LB) and E (LE); ^2^ aromatic/arginine filters by four residues, with one in helix 2 (H2) and helix 5 (H5), and two in loop E (LE1,LE2); and ^3^ five key positions in the protein sequence (P1–P5) associated with function and specific physicochemical properties for each subgroup.

The motifs in InPIPs are highly conserved: all InPIPs carry two NPA motifs; the ar/R filter is F-H-T-R in all InPIPs; and the Froger’s position is M-S-A-F-W in InPIP1s and A-S-A-F-W in InPIP2s (Table 2, Appendix A). Likewise, InTIPs have two highly conserved NPA motifs (Table 2, Appendix A); the ar/R filter is H-I/V-A/G-V/R/Y in all InTIPs (Table 2). Additionally, the Froger’s position is T/L-S/A-A/A/S-Y-W in InTIPs (Table 2).

In contrast, InNIPs carry diverse NPA motifs, ar/R filters, and Froger’s position (Table 2). The first NPA motif in InNIP5;1, InNIP5;2, InNIP5;3, and InNIP7;1 is NPS; additionally, the second NPA motif in InNIP5;1, InNIP5;2, InNIP5;3, and InNIP6;1 is NPV (Table 2, Appendix A). InNIP2;3 lacks the C-terminal region, which includes the second NPA motif, ar/R filter (except H2 and H5), and Froger’s position (except P1) (Table 2, Appendix A). NIPs can be classified into three subgroups according to their ar/R filter sequence: NIPI, NIPII, and NIPIII [2]. The ar/R filter is W-V-A-R for NIP1s, NIP3s, and NIP4s and W-V/S-A-R for InNIP1s, InNIP3;1, and InNIP4s (Table 2, Appendix A); therefore, these were classified as NIPI. The ar/R filter is A/T-I/V-G/A-R for InNIP5s, InNIP6s, and InNIP7s (Table 2, Appendix A); thus, these were classified as NIPII. Except for InNIP2;3, which lacked the C-terminal amino acid sequence (Table 2, Appendix A), the ar/R filter for InNIP2s was determined to be G-I/S-G-R; hence, these were classified as NIPIII. Among the InNIPs, the Froger’s position is generally conserved to F/L-S/T-A-Y/F-L/M/I/V in NIPI, F/Y-S/T-A-Y-L/F in NIPII, and L-T-A-Y-I in NIPIII (Table 2, Appendix A).

The first NPA motif of InSIPs is N-P-/T/L, with some variations (Table 2, Appendix A). The first NPA motif of InXIPs is NP-I/T (Table 2, Appendix A). The second NPA motif extends to the NPARC motif in XIPs of other plant species [21], and all NPA motifs in InXIPs are also NPARC motifs (Appendix A). The ar/R filter and Froger’s position in InXIPs are highly conserved to I-T-A-R and V-C-A-F-W, respectively (Table 2, Appendix A).

Furthermore, the subcellular localization of morning glory AQPs was predicted using two programs, Plant-mPLoc and Wolf PSORT II. Both programs predicted that InPIPs and InXIPs would localize to the plasma membrane, which is consistent with previous experimental results [1,15]. TIPs have been reported to be localized to the vacuolar membrane [1], and our predictions using Plant-mPLoc were consistent with this report. However, the predicted localization of most TIPs using Wolf PSORT was to the plasma membrane (Table 1), which was considered to be a misprediction. Although NIPs have been reported to localize to the endoplasmic reticulum membrane and rhizobial peribacteroid membrane, most were predicted to be localized to the plasma or vacuolar membrane (Table 1). Nonetheless, the most likely localization of NIPs occurs in the plasma membrane and endoplasmic reticulum [5]. InSIPs were predicted to be localized to the plasma or vacuolar membrane. However, SIPs have been reported to be localized to intracellular membrane systems, with the endoplasmic reticulum membrane as the most likely localization candidate [22].

### 2.5. Gene Expression in Various Organs of Japanese Morning Glory

To estimate the physiological functions of InAQPs in morning glory, gene expression data, indicated by corresponding reads per kilobase million (RPKM) values, for each organ (embryo, flower, leaf, root, seed coat, and stem) were obtained from the morning glory RNA-Seq database (Table 3). As discussed previously, the sequences of XM_019346257, XM_019304577.1, and XM_019314787.1 were used as InTIP3;1, InNIP4;3, and InNIP5;2, respectively, in further analyses. However, for the gene expression data analysis, the INIL03g01865, INIL08g31098, and INIL06g37646 sequences were used because the RPKM values of the predicted INIL03g01865, INIL08g31098, and INIL06g37646 transcripts, but not the XM_019346257, XM_019304577.1, and XM_019314787.1 transcripts, were registered in this database. Most *InPIP*s and *InTIP*s showed high expression; additionally, several of these AQPs, including *InPIP1;1*, *InPIP1;2*, *InPIP1;4*, *InPIP2;1*, *InPIP2;2*, *InPIP2;9*, *InTIP1;1*, *InTIP1;3*, *InTIP1;4*, *InTIP1;6*, *InTIP2;1* and *InTIP2;2,* exhibited a constitutive expression in all the analyzed organs. Among these, *InPIP1;2* showed a particularly high expression in all the organs. In contrast, most *InNIP*s, *InSIP*s, and *InXIP*s exhibited low expression in all the analyzed organs (Table 3). The detailed gene expression patterns of morning glory AQPs, along with their predicted transport substrates, are discussed below.

## 3. Discussion

### 3.1. AQPs Have Been Highly Conserved throughout Evolution

In the present study, 44 AQPs were identified in the morning glory genome. This number observed is similar to that in other plant genomes, such as 35 found in *Arabidopsis* [11], 33 in rice [12], 41 in potato [23], 47 in tomato [13], and 51 in flax [24]. The exon–intron structures in each AQP subfamily in morning glory (Figure 3) are similar to those in other plants, including tomatoes [13] and citrus plants [25]. Therefore, the gene structure of each AQP subfamily is conserved in plants. XIPs represent a novel clade of AQPs, first described in the moss *Physcomitrella patens* [21]. Three XIPs were identified in morning glory: InXIP1;1, InXIP1;2, and InXIP1;3. Interestingly, XIPs are present in a diverse range of plants, including tomato, potato, citrus, and flax, but not *Arabidopsis* or rice [11,12,13,23,24,25]. XIPs form a clade independent of other subfamilies and have been proposed to play specific roles in substrate transport. In morning glory, InXIPs formed an independent clade from other subfamilies (Figure 2). A phylogenetic analysis of morning glory, tomato, potato, tobacco, cotton, and citrus XIPs revealed that XIPs in morning glory are similar to those of plants belonging to Solanaceae (tomato, potato, and tobacco), which is closely related to the Convolvulaceae, to which morning glory belongs (Figure 4).

### 3.2. Prediction of AQP Transport Substrates in Morning Glory

AQPs possess several characteristic conserved motifs, including two highly conserved NPA motifs, the ar/R filter (H2, H5, LE1, and LE2), and the Froger’s position (P1–P5) (Figure 1). These motifs contribute to the selective transport of water and other substrates via AQPs. Based on the similarities of motif sequences between morning glory (Table 2) and other plants AQPs, substrates for which were identified (Appendix A), and substrates for morning glory AQPs were predicted.

PIPs are classified into PIP1 and PIP2 subgroups. PIP2s are characterized by a shorter N-terminus and a longer C-terminus than PIP1s [1]; these PIP characteristics were also observed in InPIPs. The phosphorylation of N-terminal serine residues in PIP2 positively regulates its water transport activity [1,5]. Recently, calcium-dependent protein kinase (CDPK) was shown to mediate this serine phosphorylation in gentian [18]. These serine residues are also conserved in InPIP2s (Appendix A), implying the occurrence of phosphorylation regulation of the water transport activity in these AQPs. In radish AQPs, amino acid residues I (PIP1s) and V (PIP2s), located after the second NPA motif in PIP1s and PIP2s, respectively, are important for the water transport activity; corresponding differences in these residues can explain the higher water transport activity of PIP2s compared to that of PIP1s [28]. In morning glory PIPs, the amino residue V was detected in all InPIP2s, while residue I was detected in all InPIP1s (Appendix A). Therefore, InPIP2s have a higher water transport activity than InPIP1s. The ar/R filter and Froger’s position of InPIP1s were the same as those of NtAQP1, which transports carbon dioxide and urea [2,29]. Therefore, InPIP1s are likely to transport carbon dioxide, urea, and water. *Arabidopsis* AtPIP2;1 and AtPIP2;4 transport hydrogen peroxide [2]. Therefore, considering their high similarly to AtPIP2;1 and AtPIP2;4 (Appendix A), InPIP2s are likely to transport hydrogen peroxide.

TIPs transport water, ammonia, hydrogen peroxide, and urea [1,2]. InTIP1s and InTIP2s showed high similarity to *Arabidopsis* AtTIP1;1, AtTIP1;2, and AtTIP2;3, which transport hydrogen peroxide (Appendix A). Therefore, InTIP1s and InTIP2s may transport hydrogen peroxide. The ar/R filters and Froger’s positions of the ammonia-transporting TIPs are H-I-G-R and T-S-A-Y-W, respectively [2]. The ar/R filters and Froger’s positions of InTIP2;1 and InTIP2;2 were the same (Table 2), suggesting that InTIP2;1 and InTIP2;2 transport ammonia. For TIPs that transport urea, the ar/R filters are H-I-A-V or H-I-A-R and the Froger’s positions are T-S/A-A-Y-W and T-S-A-Y-W [2]. The ar/R filters and Froger’s positions of InTIP1;1, InTIP1;2, InTIP1;4, InTIP1;5, InTIP1;6, and InTIP4;1 were consistent with these urea-associated sequences (Table 2, Appendix A). Therefore, these AQPs are predicted to transport urea. In the ar/R filters of InTIP1;1, InTIP1;2, InTIP1;4, InTIP1;5, and InTIP1;6, an amino acid residue at LE2 is V, which has a smaller molecular size than R. Thus, their pore size may be wider, and these InTIPs may be able to transport urea, which has a molecular size larger than that of water [2].

The ar/R filter of InTIP5;1 differed from that of the other InTIPs (Table 2, Appendix A). This is also the case for tomato [13] and potato [23] TIPs. Meanwhile, in *Arabidopsis*, AtTIP5;1 is highly expressed in pollen and has been reported to transport urea and water [30]. Furthermore, AtTIP5;1 is expected to transport boric acid, because the expression of *AtTIP5;1* was induced under high boron conditions, and *AtTIP5;1* overexpression enhanced high boron tolerance [31]. InTIP5;1 is highly similar to AtTIP5;1 (Appendix A) and may, therefore, transport urea and boric acid, in addition to water.

NIPs transport boric acid, urea, glycerol, formamide, and silicic acid [1,2], suggesting that InNIPs transport ammonia, glycerol, and formamide. NIPII transports glycerol, urea, and boric acid [2]. *Arabidopsis* AtNIP5;1, AtNIP6;1, and AtNIP7;1 contribute to boron transport [32,33,34]. Expanded NPA motifs of InNIP5;1, InNIP5;2, InNIP5;3, InNIP6;1, and InNIP7;1 were NPV or NPS, similar to those of AtNIP5;1, AtNIP6;1, and AtNIP7;1. Furthermore, the ar/R filters and Froger’s positions of InNIP5;1, InNIP5;2, InNIP5;3, InNIP6;1, and InNIP7;1 were similar to those of AtNIP5;1, AtNIP6;1, and AtNIP7;1. Therefore, InNIP5;1, InNIP5;2, InNIP5;3, InNIP6;1, and InNIP7;1, which are highly similar to AtNIPs (Appendix A), may also transport boric acid.

Urea, which has a molecular size similar to that of boric acid, is also transported by NIPII [2]; therefore, InNIP5;1, InNIP5;2, InNIP5;3, InNIP6;1, and InNIP7;1 may transport urea. NIPIII transports urea, boric acid, and silicates [2]. The ar/R filters and Froger’s positions of silicic acid transporting NIPs are G-S-G-R and L-T-A-Y-F, respectively [2], whereas the Froger’s position of InNIP2;2 differs slightly (Table 2, Appendix A). Therefore, it is unclear whether InNIP2;2 transports silicic acid.

AtSIPs are localized to the endoplasmic reticulum membrane; moreover, while AtSIP1;1 and AtSIP1;2 transport water, AtSIP2;1 did not confer a water transporting function in a yeast heterologous expression system [22]. Therefore, whether InSIPs possess water transport activity remains unclear.

The first three amino acid residues (H2, H5, and LE1) in the ar/R filter of InXIPs are hydrophobic amino acids (Table 2, Appendix A); thus, InXIPs may contribute to the transport of molecules other than water [2]. Solanales XIPs transport hydrogen peroxide, glycerol, urea, and boric acid, as evidenced by the measurements of transport activity using *Xenopus laevis* oocytes or a yeast heterologous expression system [15]. InXIPs are highly similar to Solanales XIPs (Figure 4); thus, InXIPs may transport hydrogen peroxide, glycerol, urea, and boric acid.

### 3.3. Physiological Roles of AQPs in Morning Glory

In plants, AQPs maintain water homeostasis and play diverse roles in stress response, cell elongation and expansion, stomatal opening and closing, flower opening and closing, fertilization, germination, and fruit enlargement [1,19,35]. The physiological functions of AQPs in morning glory are discussed below based on organ-specific gene expression patterns and predicted substrates.

PIPs and TIPs have been reported to possess a potent water transport capacity [1]. *InPIP1;2* was expressed in all analyzed organs, and its expression level (RPKM value) was higher than that of the other AQPs (Table 3). Therefore, InPIP1;2 may be the major AQP in morning glory. Further, the expression of *InPIP1;2* was particularly high in roots. Additionally, *InPIP1;3*, *InPIP2;6*, *InPIP2;7*, *InPIP2;9*, *InTIP1;1*, *InTIP1;4*, *InTIP1;6*, *InTIP2;1*, and *InTIP4;1* were also highly expressed in roots (Table 3). Therefore, these AQPs may play important roles in water uptake from the soil to the roots. Wang et al. (2020) [35] reported that PIPs and TIPs were abundant in roots and were important for root growth, suggesting that InAQPs also play an important role in root growth.

*InPIP1;2*, *InPIP2;3*, *InTIP1;3*, and *InTIP2;1* were highly expressed in leaves (Table 3); thus, these AQPs are expected to contribute to water transport in leaves. Additionally, some PIPs transport carbon dioxide [29]. In tobacco, the overexpression of *NtAQP1*, which transports carbon dioxide, increased photosynthetic activity and promoted leaf growth [35]. Therefore, InPIP1;2 and InPIP2;3, which are highly similar to NtAQP, may play vital roles in photosynthesis through carbon dioxide transport.

In the current study, *InPIP1;1*, *InPIP1;2*, *InPIP1;4*, *InPIP2;1*, *InPIP2;9*, *InTIP1;3*, *InTIP1;4*, and *InTIP2;1* were highly expressed in stems (Table 3). *Arabidopsis AtTIP1;1* is highly expressed in the hypocotyl, and its expression pattern is related to cell elongation [36]. In rice, *PIP* and *TIP* transcripts are concentrated at the internodes of growing plants [37]. Therefore, InAQPs contribute to stem elongation.

*InPIP1;1*, *InPIP1;2*, *InPIP2;1*, *InPIP2;9*, *InTIP1;3*, *InTIP1;4*, and *InTIP2;1* were highly expressed in flowers (Table 3). AQPs have been reported to contribute to the opening and closing of flowers in several plants, including Japanese morning glory [1], tulips [17], roses [38,39], and barley [19]. In tulips, *TgPIP1;1*, *TgPIP1;2*, *TgPIP2;1*, and *TgPIP2;2* were expressed in petals; in particular, *TgPIP2;2* was highly expressed, indicating its significance to flower opening within this plant [17]. In roses, *RhPIP2;1* and *RhTIP1;1* were involved in flower opening [38,39]. Additionally, in barley, it was suggested that *HvTIP1;1*, *HvTIP1;2*, *HvTIP2;3* and *HvPIP2;1* are important in the flowering process [19]. In Japanese morning glory, flower opening occurs within a few hours and it requires abundant and rapid water uptake into petal cells and their vacuoles. Therefore, InPIP1;1, InPIP1;2, InPIP2;1, InPIP2;9, InTIP1;3, InTIP1;4, and InTIP2;1 may play critical roles in the uptake of water into petal cells and their vacuoles. In fact, mercury, an AQP inhibitor, has been reported to suppress flower opening in morning glory [1]

In tulips, PIP2 phosphorylation regulates light- and temperature-dependent flower opening and closing [40,41]. Recently, CDPK, which is involved in PIP2 phosphorylation, was identified in gentians [18]. CDPK phosphorylates serine residues at the C-terminus of GsPIP2s; this phosphorylation is essential for the light-dependent opening of gentian flowers. The phosphorylated serine residue at the C-terminus is conserved in InPIP2s, including InPIP2;1 and InPIP2;9 (Appendix A). Thus, the phosphorylation of InPIP2s, including InPIP2;1 and InPIP2;9, may be important for the opening of Japanese morning glory flowers. Further, using an anti-phosphorylated serine antibody, which was used in the previous study on gentian [18], we confirmed the phosphorylation of the serine residue at the C-terminus of InPIP2s during the opening of the Japanese morning glory flowers (unpublished). Therefore, InPIP2 phosphorylation is also important for flower opening in Japanese morning glory.

*InPIP1;2*, *InPIP1;4*, *InPIP2;1* and *InPIP2;9* were highly expressed in the seed coat, whereas *InPIP1;2* and *InTIP3;1* were highly expressed in the embryos (Table 3). These AQPs may be involved in the water uptake or dehydration in seed coats or embryos during the seed coat’s maturation or germination. *InTIP3;1* was specifically expressed in embryos (Table 3). TIP3 was originally identified as αTIP—a dominant TIP in the storage vacuoles of seed coats [42]. In *Vicia faba*, TIP3 promotes water uptake at the early stages of germination [35]. Therefore, InTIP3;1 may be involved in the water uptake during the seed germination of morning glory.

Overall, the expression levels of *InNIP*s in Japanese morning glory were low, which is consistent with findings in other plant species [24,43]. Nonetheless, *InNIP1;1* was expressed in all organs, *InNIP2;1* was expressed in leaves, and *InNIP5;1* was expressed in roots (Table 3). InNIP5;1 may contribute to boric acid and urea uptake in the roots. In fact, the homolog of InNIP5;1 in *Arabidopsis*, AtNIP5;1, has been reported to be involved in the uptake of boric acid in roots [44].

*InSIP1;1* was expressed in all organs, albeit at low levels. However, *InSIP2;1* was not expressed in any organ (Table 3). No or low expression of *InXIP*s was detected in the analyzed organs. Low expression levels of *InXIP1;2* in leaves and very low expression levels of *InXIP1;3* in stems were detected (Table 3). These results are consistent with the low expression of *XIP*s observed in other plants [43,45]. The expression levels of some genes with low initial expression are altered depending on the growth stage and external environment. Therefore, gene expression should be further analyzed at different growth stages and under various environmental conditions.

## 4. Materials and Methods

### 4.1. Identification of AQPs in Japanese Morning Glory

Initially, a BLASTP search of the Japanese morning glory genome (http://viewer.shigen.info/asagao/download.php (accessed on 30 January 2021)) was performed using the AQP amino acid sequences from *Arabidopsis* [11] and tomato [13], as queries under the default setting (e-value: 1 × e^−10^). Among the candidate Japanese morning glory AQPs, those with fewer than 100 or more than 350 amino acid residues were excluded. Moreover, candidates without an expanded NPA motif (NPA/I/L/S/T/V) were excluded. Gene structure, specifically the structure of the untranslated regions, exons, and introns, was visualized using the Gene Structure Display Server 2.0 (http://gsds.gao-lab.org/ (accessed on 22 August 2022)).

### 4.2. Multiple Sequence Alignment and Phylogenetic Analysis

Multiple sequence alignments of amino acid sequences were performed using CLUSTALW (https://www.genome.jp/tools-bin/clustalw (accessed on 19 October 2021)) with the default settings. Phylogenetic trees were constructed using the neighbor-joining algorithm in CLUSTALW and visualized using MEGAX [46].

### 4.3. Prediction of TMD, Molecular Weight, Isoelectric Point (pI), and Subcellular Localization

TMDs were predicted using the TMHMM Server v.2.0 (http://www.cbs.dtu.dk/services/TMHMM/ (accessed on 29 April 2021)) and SOSUI (https://harrier.nagahama-i-bio.ac.jp/sosui/sosui_submit.html (accessed on 29 April 2021)). The molecular weight (kDa) and pI of proteins were predicted using ProtParam (http://web.expasy.org/protparam (accessed on 29 April 2021)). Subcellular localization was predicted using Plant-mPLoc (http://www.csbio.sjtu.edu.cn/bioinf/plant-multi/ (accessed on 28 May 2021)) and WoLF PSORT II (https://www.genscript.com/wolf-psort.html (accessed on 28 May 2021)).

### 4.4. Prediction of Gene Expression Profile

Gene expression data, expressed as RPKM values for different organs and tissues of Japanese morning glory, were obtained from the Japanese morning glory RNA-Seq database (http://viewer.shigen.info/asagao/jbrowse.php?data=data/Asagao_1.2 (accessed on 30 October 2021)). Details of the samples used for expression analysis have been reported previously (Table S21 in Hoshino et al., 2016 [16]).

## 5. Conclusions

The present study offers a comprehensive overview of AQPs in morning glory. A total of 44 AQPs were identified in the morning glory genome, and their details were summarized. Morning glory has been widely used as a model plant for flowering and floral organ development. In the present study, our findings suggest that morning glory AQPs are involved in flower opening. Overall, the information presented here can help clarify the physiological functions of AQPs in this plant.

## Figures and Tables

**Figure 1 plants-12-01511-f001:**
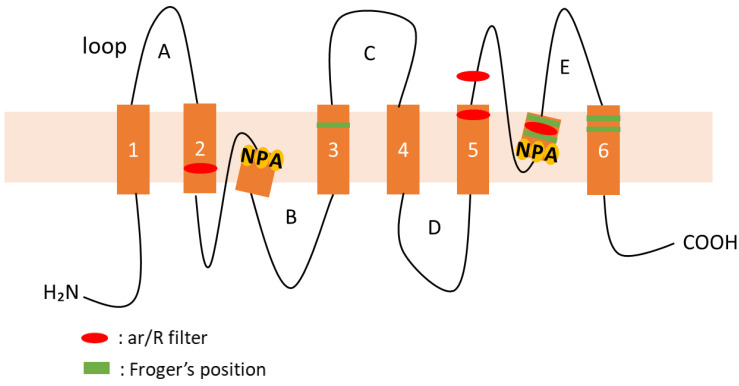
**Schematic representation of the classical structure of AQPs.** An AQP monomer showing the six transmembrane helices (1–6) connected by two intracellular (loop B and D) and three extracellular (loop A, C, and E) loops. The conserved residues from the first selective filter (NPA motifs) and the second filter (ar/R filter) are shown in yellow and red, respectively. Froger’s positions are shown in green, respectively.

**Figure 2 plants-12-01511-f002:**
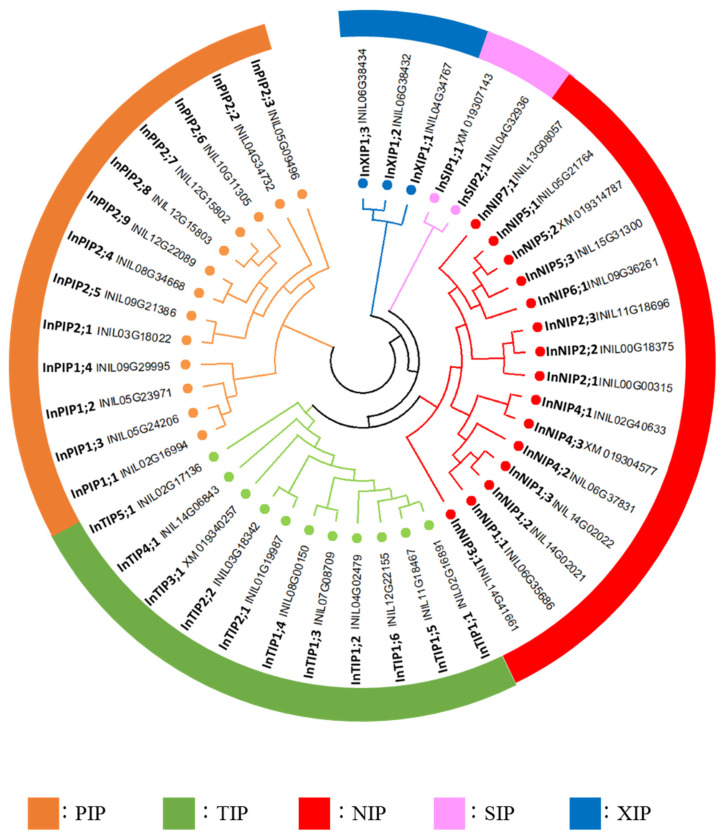
**Phylogenetic tree of the AQPs of Japanese morning glory.** Phylogenetic tree was generated by the neighbor-joining method derived from a CLUSTAL alignment of the AQP amino acid sequences of Japanese morning glory.

**Figure 3 plants-12-01511-f003:**
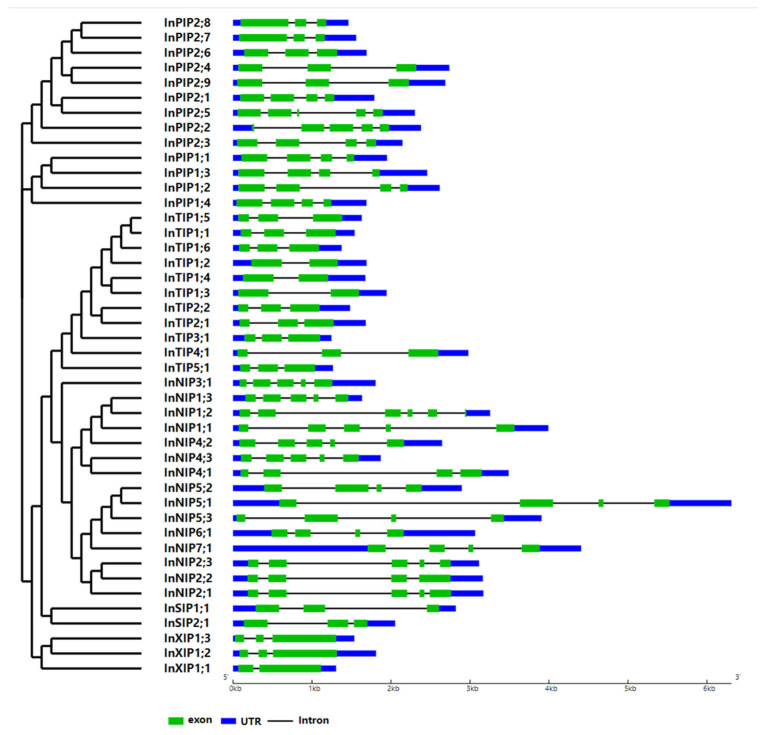
**Exon–intron structure of the AQP genes of Japanese morning glory.** A graphic representation of the gene models of all 44 AQPs identified in this study. Gene models are based on genome and CDS sequences.

**Figure 4 plants-12-01511-f004:**
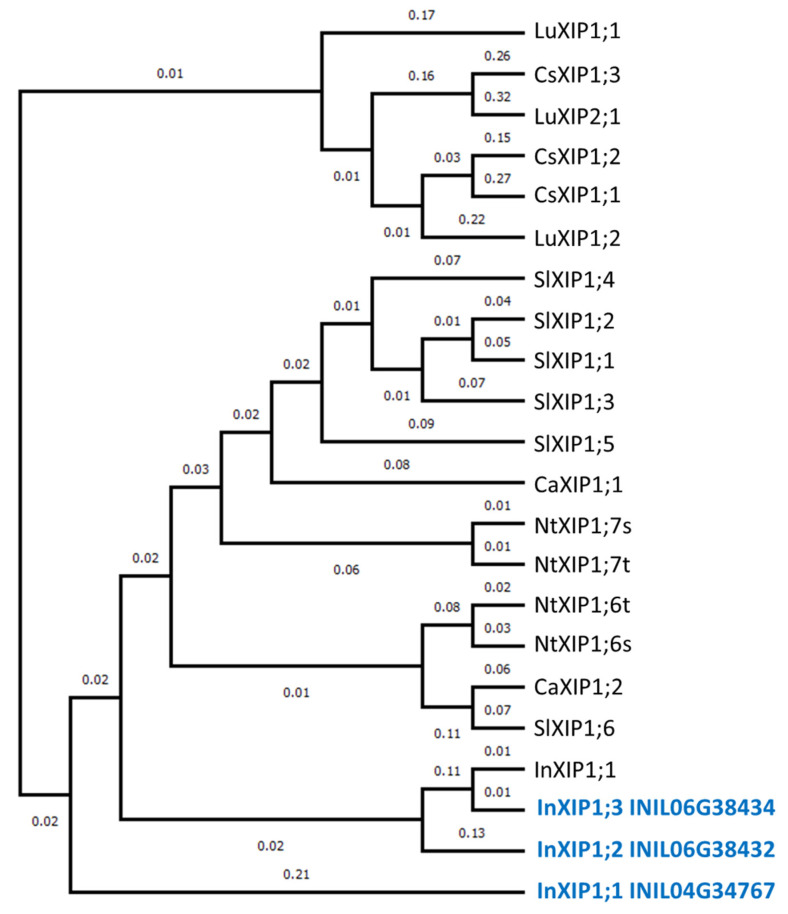
**Phylogenetic tree of XIP family members.** A phylogenetic tree was generated by the neighbor-joining method derived from a CLUSTALW alignment of the AQP amino acid sequences of Japanese morning glory InXIPs from this study (indicated in blue) and InXIP1;1 (HM475296) [15], alongside tobacco NtXIPs [26], tomato SlXIPs [13], pepper CaXIPs [27], citrus CsXIPs [25], and flax LuXIPs [24]. Numbers on the phylogenetic tree indicate branch length.

**Table 3 plants-12-01511-t003:** **Gene expression of AQPs in Japanese morning glory.**

Name	Embryo	Flower	Leaf	Root	Seed Coat	Stem
*InPIP1;1*	33.9	362.3	83.8	201.8	82.7	475.8
*InPIP1;2*	637.8	2494.5	1009.1	6100	1457.3	2145.3
*InPIP1;3*	22	109.2	93.1	741.9	1.4	65.8
*InPIP1;4*	13.7	89.2	30.1	223.1	1059.5	359.4
*InPIP2;1*	82.9	683.8	130.7	170.4	1744.5	857.8
*InPIP2;2*	192.7	107.4	18.1	8.4	54	125.7
*InPIP2;3*	13	35	985.5	0	74.2	90.5
*InPIP2;4*	0.1	38.8	13.9	254.7	104.8	195
*InPIP2;5*	0	0	0	0	0.6	0
*InPIP2;6*	0	0.7	0.5	389.9	0	1.1
*InPIP2;7*	1	117.8	50.3	386.7	6.8	127.7
*InPIP2;8*	0.9	0	68.5	0	0	86.3
*InPIP2;9*	33	329.7	200.5	1310.8	511.9	1028.8
*InTIP1;1*	2.7	50.6	46.1	345.3	3	22.3
*InTIP1;2*	0.1	0	59.7	17.7	16.8	0
*InTIP1;3*	10.8	450.1	1248.7	191.4	92.5	817.5
*InTIP1;4*	5	851.8	108.2	572.8	143	777.8
*InTIP1;5*	0.3	39.9	2.9	1.1	0	4.1
*InTIP1;6*	0.4	4.2	0.6	938.7	12.8	0.6
*InTIP2;1*	18.2	601.6	864.8	1525.7	173.6	596
*InTIP2;2*	0.2	1.6	0.7	155.7	117	112.8
*InTIP3;1*	1920.8	0	0.2	0.1	0	0
*InTIP4;1*	0	25.7	9.7	297.4	6.6	106.8
*InTIP5;1*	0	1.7	0	0	0	0.1
*InNIP1;1*	20.4	38.4	24.4	20.6	23	41.3
*InNIP1;2*	0	0.3	0.7	0	0	0
*InNIP1;3*	0	0.5	0.9	0	0	0
*InNIP2;1*	0	7.6	18.5	2.3	0.1	7.7
*InNIP2;2*	0	4.2	10.1	1.1	0	3.3
*InNIP2;3*	0	2.6	5.1	0.5	0	1.7
*InNIP3;1*	0	0	0	0.3	0	0
*InNIP4;1*	0	0	0.2	0.2	0.6	0.6
*InNIP4;2*	0	0	0.3	0	0	0.8
*InNIP4;3*	0	0	0	0	0	0.2
*InNIP5;1*	4	2	0.7	95.9	0.1	18.6
*InNIP5;2*	0	0	0	0	0	0
*InNIP5;3*	0	0	0	0.2	0	0.3
*InNIP6;1*	0.3	0	0	0	0	0
*InNIP7;1*	0	0	0	0	0	0
*InSIP1;1*	26.4	58.5	17.5	48	88.9	53.7
*InSIP2;1*	0	0	0	0	0	0
*InXIP1;1*	0	0	0.7	0	0	0
*InXIP1;2*	0	0	17.6	0.1	0	0
*InXIP1;3*	0.5	0	0	0	0	1.8

Gene expression profile data in Japanese morning glory organs was obtained from the Japanese morning glory RNA-Seq database (http://viewer.shigen.info/asagao/jbrowse.php?data=data/Asagao_1.2 (accessed on 30 October 2021)). Numbers in the table indicate RPKM values. Gene expression levels (low to high) are indicated by white to deep-red color shades. Data for InSIP1;1 are shown as RPKM values for Cufflinks.

## Data Availability

Data can be requested from the corresponding authors.

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
