# Peer review of "Genome-Wide Analysis of Aquaporins in Japanese Morning Glory (Ipomoea nil)"

_plants, 2023, doi:10.3390/plants12071511_

Round 1

Reviewer 1 Report (Previous Reviewer 3)

The manuscript has been significantly improved and can be accepted in its present form.

Author Response

Your comment:The manuscript has been significantly improved and can be accepted in its present form.

Response: We are happy to hear that you mentioned this manuscript is worthy of publication in Plants. We hope our manuscript will be accepted by Plants and published near future.

Reviewer 2 Report (New Reviewer)

(1) The writing of the manuscript needs improvement, and some grammatical and spelling errors still exist. Suggest to invite someone native to help with the polish.

(2) In the Abstract, the authors presented too many introductions on the AQP background, while less on the results, and please re-write this portion.

(3) Some Latin names were not italic, and check the whole manuscript.

(4) Most of the Tables or Figures in this manuscript were presented across two pages, and the authors should think about the legibility and aesthetics of this article.

(5) All the results were provided by bioinformation analysis, while there were no experiments to verify the accuracy. Suggest to add the qRT-PCR experiments.

Author Response

Thank you for reviewing our manuscript. We are very grateful for your insightful advice and comments to improve our manuscript. We modified our manuscript according to your advice except for 5th comment. Below are our point-by-point responses.

Comments to the Author

  1. The writing of the manuscript needs improvement, and some grammatical and spelling errors still exist. Suggest to invite someone native to help with the polish.

Response: Thank you for your helpful suggestion. We again conducted an English editing by a native English speaker.

  1. In the Abstract, the authors presented too many introductions on the AQP background, while less on the results, and please re-write this portion.

Response: Thank you for the helpful suggestion. We agree with you. We rewrote it with more results. Please check lines 27 through 33.

  1. Some Latin names were not italic, and check the whole manuscript.

Response: Thank you for the helpful suggestion. We agree with you. We rechecked the entire manuscript and have replaced them as you suggested.

  1. Most of the Tables or Figures in this manuscript were presented across two pages, and the authors should think about the legibility and aesthetics of this article.

Response: Thank you for your helpful suggestion. We agree with you. However, the author’s instruction says that text size should be at least 8 point, therefore the tables are presented across multiple pages. We will discuss about this with the publisher after aceptance.

  1. All the results were provided by bioinformation analysis, while there were no experiments to verify the accuracy. Suggest to add the qRT-PCR experiments.

Response: Thank you for your helpful suggestion. We also think it is better to add qRT-PCR data. However, the concept of this study is to summarize information on AQPs in Japanese morning glory based on the public genome and transcriptome database. Further studies, including qRT-PCR, are left to other morning glory researchers and AQP researchers.

Reviewer 3 Report (New Reviewer)

In this paper, Tamami Inden et al, systematically identified 44 AQPs in morning glory and predicted the physiological functions of AQPs. The experimental work has been clearly presented by the authors, which provides new insights of morning glory AQPs in flowering. But some minor points need to be addressed before acceptance.

Minor points:

1. Latin names of plant species should be italicized. For example, on line 90, line 102, and line 298, “Arabidopsis” should be “Arabidopsis”.

2. The article format needs to be more standardized. The caption font of Figure 3 does not need to be bold.

3. The grammar of written English needs to be improved. For example, on line 216 “the sequence of” should be “the sequences of”. On line 218 “were used” should be “was used”. The authors should carefully go through the whole manuscript to rectify any grammatical mistakes.

Author Response

Thank you for reviewing our manuscript. We are happy that you mentioned our manuscript is novelty. We are very grateful for your helpful advice and comments to improve our manuscript. Below are our point-by-point responses.

Comments to the Author

  1. Latin names of plant species should be italicized. For example, on line 90, line 102, and line 298, “Arabidopsis” should be “Arabidopsis”.

Response: Thank you for the helpful suggestion. We checked the entire manuscript and have made italic as your suggestion.

  1. The article format needs to be more standardized. The caption font of Figure 3 does not need to be bold.

Response: Thank you for your comment. We have replaced bold with fine font.

  1. The grammar of written English needs to be improved. For example, on line 216 “the sequence of” should be “the sequences of”. On line 218 “were used” should be “was used”. The authors should carefully go through the whole manuscript to rectify any grammatical mistakes.

Response: Thank you for your helpful suggestion. We again asked a professional English editor for English editing. Please look at the certification of English editing.

Round 2

Reviewer 2 Report (New Reviewer)

All the incorrect portions have been modified, and we believe the revised manuscript meet the criteria for publication on the top journal of Plants.

This manuscript is a resubmission of an earlier submission. The following is a list of the peer review reports and author responses from that submission.

Round 1

Reviewer 1 Report

The content of line 9/10 should be controlled; not at the right position in the Abstact. 

This MS is well able to make a further contribution to expanding molecular libraries and contributing to the understanding of aquaporins in plants. Therefore it is suitable for a Special Issue in "Bioinformatics".

Reviewer 2 Report

The authors identified AQPs in Ipomoea nil using the Japanese morning glory genome database. They schematized the gene structures based on their genome and coding sequences. Additionally, they predicted the TMDs (transmembrane domains), several conserved and characteristic protein motifs and subcellular localization of morning glory AQPs.

Moreover, using this identification and gene expression data analysis of different organs, embryo, flower, leaf, root, seed coat, and stem and the previously known functions in other plant species they estimated the putative roles of InAQPs.

Additional comments:

Abstract: line 9: you can remove “also”

line 21: “amino acid motifs harboring permeability pores.” Don’t sound good to me

line 54, 62, 78: lack of one space

line 79: replace “understand” by “predict”

line 82: the putative roles

line 109-123: Maybe it is better to explain this topic in the chapter - 2.5. Gene expression in various organs of Japanese morning glory

line 164: These motifs are important for the selectivity of substrates transport [2,6].

line 201: extra paragraph

Reviewer 3 Report

In the article “Genome-wide analysis of aquaporins in Japanese morning 2 glory (Ipomoea nil)”, authors have identified AQPs in morning glory (Ipomoea nil) based on homology search and further characterized the structural motifs, subcellular localization, and tissue-specific expression based on available RNAseq dataset. The article is informative, however,

The abstract and result sections are not well written. For example, abstract starts with “AQPs also transport low-molecular-weight solutes, including boric acid, 9 glycerol, urea, and ammonia.”

The beginning of the result section is unclear and it makes it difficult to follow the results.

Please include a table to show the percent homology, query coverage, and e-value of the BLASTP search with Arabidopsis, tomato homologs

Since every species have its specific addition and deletion, it is not clear why authors choose to use XM_019346257.1, XM_019304577.1, and XM_019314787.1 for further analysis, as the species-specific sequence would have provided better information.

In table 3, are genes having no RPKM have stress-regulated expression? Is there stress induced RNAseq dataset for  Ipomoea nil to check it?